# In My New Home: The Daily Lives of People Living in Public Houses after a Long Period of Homelessness

**DOI:** 10.3390/bs12110416

**Published:** 2022-10-27

**Authors:** Marta Gaboardi, Chiara Bonechi, Eleonora Zamuner, Massimo Santinello

**Affiliations:** Department of Developmental Psychology and Socialization, University of Padova, 35131 Padova, Italy

**Keywords:** homelessness, relationships, housing, interviews, community

## Abstract

Homelessness refers to a loss of social relationships and a condition of isolation and stigma that affects a person’s well-being. Although the literature has revealed the crucial role of a home in a person’s well-being, few studies have explored the daily lives of people who transition from homeless services to an independent home. People who experience homelessness are at risk of remaining connected to homeless services even after finding a home. This study aimed to explore the daily lives of people who have obtained public housing, focusing on their daily relationships and the places they frequent. Data were collected through interviews with quantitative and qualitative measures involving 14 people with a history of homelessness who had obtained a public house in a medium-sized Italian city. Several themes concerning social relationships and places were identified. Regarding social relationships, people experience loneliness or a connection with the community and homeless services. They spend their time alone at home or around the city. The implications of the results are discussed with respect to practice and research on homelessness.

## 1. Introduction

Having a home ensures stability, privacy, independence, and a sense of control, which provides a secure base for identity construction [1,2]. Additionally, having developed a sense of belonging to a particular place, such as a home, is associated with greater subjective psychological well-being [3]. The psychological perception of home refers to the feeling of belonging to others. Psychological home implies different aspects, such as the ideas that people have about themselves in relation to a home, a sense of security and protection, and placing attachments in the physical environment to support self-identity [4,5]. Moreover, the construct of a psychological home is related to a sense of community [4,6], as the concept of home may be extended to the neighbourhood and community.

These aspects are particularly important for people who have been homeless for long periods of time and are transitioning from being homeless to having a home. Homelessness affects people physically, psychologically, and socially. People experiencing homelessness have greater mental and physical health problems [7] and lower life expectancy [8] than the general population and often experience loneliness and social stigma [9]. People with a history of homelessness identify home as the place where they can build their daily routines, control their own lives, think about their future, and cultivate social relationships [2,10]. A home provides people with a place where they can be free and construct meaningful aspects of their lives [11]. In a home, people can experience freedom, privacy, safety, and emotional connections [12].

Homeless services (such as shelters and transitional homes) hinder housing stability by increasing users’ dependence on services and even acclimating them to an institutionalised life [13]. Moreover, traditional services are focused on people’s basic needs in terms of food, health, and finding temporary housing, with the risk of creating an ‘institutional circuit’ of streets and homeless services [14]. Permanent supportive housing programs can foster well-being and housing stability [15,16,17].

Despite the evidence supporting the importance of housing, housing alone is not always sufficient to address the issues associated with homelessness, such as social exclusion [18]. Although people start living in independent houses, social exclusion, loneliness, and lack of support can persist [19]. The detrimental effect of social isolation may remain regardless of housing [18,19,20] because they may still encounter different challenges after obtaining housing, limiting their integration. Factors such as isolation, loneliness, unemployment, lack of social support, and psychiatric issues are some possible barriers to community integration [20]. People may experience ‘not-fitting-in’ the neighbourhood, community, and wider society [21]. Moreover, the stigma related to homelessness, which comprises poverty, mental health issues, and disconnection from others, contributes to social exclusion, discrimination, and low integration level [22].

Overall, even after gaining housing stability, people struggle to rebuild their place in society and feel a sense of inclusion [22]. Therefore, community integration plays an important role in promoting a sense of belonging to wider society and enhancing people’s well-being [19,23]. Community integration, as a multidimensional concept, includes physical integration, referring to participation in daily community activities; social integration, referring to social contact with others and social support; and psychological integration, referring to one’s sense of community and belonging [23]. Particularly, along with housing stability, social support is a predictor of overcoming homelessness [18], promoting mental well-being [22], and decreasing the sense of loneliness [11]. Social support represents the possibility of breaking away from homeless services and participating in community life [24]. From their perspective, people experiencing homelessness feel integrated when they perceive a sense of personal dignity, respect from others, and a sense of usefulness within society [25].

Therefore, even when people obtain housing after a long period of homelessness, they must recreate a psychological sense of connection with a new home, others, and their community. Thus, to understand how people experience these aspects, the present research aimed to explore the daily lives of people with a history of homelessness living in public houses, focusing on their social relationships and the places they frequent.

## 2. Materials and Methods

### 2.1. Context

In 2021, an increasing number of people experiencing homelessness in Padua received public housing allocations through a municipal notice for public housing allocation. The municipality of Padua published its annual call for public housing allocation. Housing is allocated on the basis of the score obtained in the ranking list and compliance with specific requirements (e.g., low income). In 2021, 27 people experiencing homelessness won the ranking list and became homeowners. Padua City decided to invest in a ‘Support to Housing’ project with local organisations that previously helped those people in homeless services such as shelters and group homes. Our research team was involved in the project to monitor and evaluate the activities. Before starting the research, meetings with social service providers were conducted by two research authors to share the evaluation strategy. The research comprised interviewing the participants using self-report questionnaires and open-ended questions. The results of the instruments that explore the daily lives of the participants are the focus of this article.

### 2.2. Procedure

The present study was conducted from October 2021 to March 2022. A researcher with a social service provider (different for each participant) conducted the interviews. Before proceeding, the participants signed an informed consent form and voluntarily participated in the study. Interviews lasted an average of two hours and were audio recorded.

For this study, three closed-ended and three open-ended questions were included. The three closed-ended questions asked were ‘Which people do you meet most during the week?’; ‘Which people do you do something together with?’; ‘Which people would you ask for help if you needed it?’ People can choose more than one option from the following: friends, family members, partners/intimate relationships, acquaintances, services, people known in homeless services, and neighbours. The open-ended questions included ‘How do you spend your days?’ and ‘What places and areas of the city do you visit most?’

### 2.3. Participants

Overall, 14 participants consented to participate in the present study. Amongst these, 2 were women and 12 were men with a mean age of 56.4 (range 33–83, SD = 13.57); three were foreigners (non-EU). Participants had experienced homelessness for 10 years (range: 4–27, SD = 6.5). Before housing, seven lived in a public shelter, five in a group home, and two in a street or an abandoned house. According to the European Typology of Homelessness and Housing Exclusion (ETHOS) [26], participants were roofless (without a shelter of any kind, sleeping rough) or houseless (with a place to sleep, but temporary in institutions or shelters). At the time of the study, people had lived in their house for 1 to a maximum of 11 months.

### 2.4. Data Analysis

For quantitative data, frequencies were calculated using SPSS software 28.0. The qualitative data were analysed by two master’s students using thematic analysis [27] on the basis of the following steps: (a) familiarising with the data, (b) generating initial codes, (c) searching and defining themes, and (d) reviewing themes. Researchers began by transcribing the interviews and reading them. Then, they separately developed codes at the sentence level for all interviews. Together, they collected codes into themes and discussed them with a third researcher (a postdoctoral researcher) who helped reach a consensus. Finally, the three researchers synthesised themes, codes, and phrases related to them. In the Results section, the quotes are reported in English. To ensure anonymity, participants’ interview excerpts were identified with participant identification numbers (ID numbers), along with the number of months the participant had been living in the new apartment.

## 3. Results

As summarised in Figure 1, most people meet acquaintances or people from services. When asked about who they spend most of their time with during the day, most participants answered people from services (or friends), the same individuals from whom they seek assistance.

Figure 2 depicts the themes related to social relationships and places that were interconnected.

### 3.1. Social Relationships

The theme ‘relationships’ refers to each activity involving others during the daily life of participants and comprises relationships within the community and relationships within homeless services. An additional dimension that has been identified is ‘feeling alone’, which can be conceptualised as loneliness or the absence of relationships. The subthemes are presented below.

#### 3.1.1. Relationships within a Community

Relationships within a community refer to family, friends, and acquaintances. Some participants reported keeping in touch with friends or their families:

‘*However this is my sociality, some phone calls, sometimes there is a friend of mine, a couple of friends who have remained true to me*’ (ID 5, 5 months); ‘*In the afternoon if I don’t have things to do outside, I see some friends*’ (ID 2, 7 months); ‘*Yes like this, then there is my sister calling me*’ (ID 10, 4 months).

Participants plan a daily routine by spending time with friends or other people in the wider community, as one participant noted: ‘*A walk maybe and so maybe around 6:30 am I always meet my friends, my usual friends … it always gets better than worse*’ (ID 4, 6 months); or occasionally: ‘*Then yes in the centre, a walk in the centre if there is some free friend let’s take a walk, that’s enough… I like to go around*’ (ID 12, 11 months). Moreover, one participant enjoys sitting around and meeting new people: ‘*No nothing, I sit there on the table, a small table, I eat one… whatever I like, I drink an aperitif and then I say goodbye to everyone and leave*’ (ID 13, 5 months).

#### 3.1.2. Relationships in the Homeless Services

Some participants maintained connections with the people they met in homeless services such as shelters, day centres, and services distributing food. Despite housing, participants continued to seek help from homeless services. Some people continue to use services to keep in touch with social service providers. One participant still feels safe and protected knowing that someone is there to take care of her: ‘*I am very happy if someone takes care of me, I have some company, I have something, otherwise I will go there alone*’ (ID 10, 4 months). Concurrently, engaging in homeless services helps them determine whether job opportunities exist in the city. One participant declared: ‘*Yes, I go there to greet them, see if there is any news, a project or something, that’s it*’ (ID 3, 5 months).

Cultivating connections with homeless services is useful for participants to receive assistance with housing: ‘*[Name of social provider] is the one who helped me move without my asking*’ (ID 5, 5 months); ‘*I went with them (social service providers), we got the quilt then I also asked for some advice*’ (ID 10, 4 months).

Some participants said that they usually hang out at some homeless services (day centres and services distributing food) for relational reasons instead of going there to satisfy more practical needs. They had good memories of social service providers, developed strong and trusting relationships with them, and desired to remain in touch. One participant claimed that he occasionally visited the Caritas offices to ask for job search support and to greet some social service providers whom he now considers friends:

‘*So where do I start, I usually start from Caritas, which I know and where I have some friends (amongst the social service providers), they told me, you just have to call and say (name of the user), I need help, so they know me well […]*’ (ID 3, 5 months).

Given the presence of social service providers, homeless services are viewed as safe havens where people can seek assistance and support. Another participant said he enjoys going to a Christian association, as he knows very well the priest who is active in the community and enjoys the possibility of spending some time in the company, as noted:

‘*For example, on Friday nights, I go to the church (Christian associations), I go to the church where there is (name of the priest), I have known him for many years, since when he was not a priest yet, now he is the priest of the church […] I go there on Friday nights because I know they are meeting (the volunteers of the community), we participate in a meeting, then there is the prayer and the function, and after that we spend some time together*’ (ID 2, 7 months).

#### 3.1.3. Feeling Alone

Some participants experienced loneliness owing to a lack of friends or many significant others in their lives for different reasons. One participant prefers quiet places with few people around to avoid sickness and mess: ‘*No, I only like parks, I am afraid of the mess, I go home quietly alone, I don’t like it so much, even now I have no friends, nothing, always alone, there is so much mess around, always alone*’ (ID 9, 8 months).

For one participant, establishing new friendships is quite difficult due to problems with the justice system: ‘*Now I don’t go to bars or places where there are a lot of people anymore, now even less because now I’m at home and I don’t know if I told you that I can’t hang out with convicts, I don’t have to go where the convicts are and stuff like that, you know?*’ (ID 6, 3 months). Another person prefers to live on his own because he can enjoy his own space: ‘(I spend my days) *as a true misanthrope*’ (ID 5, 5 months).

### 3.2. Places

The theme ‘places’ refers to how participants interact with their homes and the larger community. People report spending time at home to relax or because they have nothing better to do with their time, whereas in the larger community, participants tend to frequent homeless services (day centres and services distributing food) and local community services to meet basic needs and, in the latter case, to socialise.

#### 3.2.1. Opportunity of Staying at Home during Free Time

Some reported that they appreciated living in their own apartment as it allowed them to spend their free time at home. For some, the apartment embodies the place to go after work and where one can take a break, as a participant noted: ‘*My free time is when I come home, I put the music on so I can relax*’ (ID 7, 4 months). Additionally, an apartment is seen as a place where they can escape from daily commitments and troubles. Another participant claimed: ‘*Yes, (at home) I try to take a break and not think about my job, lately I stay home and watch movies*’ (ID 12, 11 months). Finally, a home provides a safe space during cold seasons and bad weather, as one person stated: ‘*Now the weather is bad, you don’t go out […] in the afternoon I do not go out unless I have something to do, otherwise I stay home*’ (ID 9, 8 months).

#### 3.2.2. Staying at Home Because There Is Nothing Else to Do

Some participants reported that they spend all their days at home, without working or exploring their neighbourhood or the city, as one person said: ‘*Interviewer: […] Are you always in your house or do you go around in the city centre, in your neighbourhood? ID 6: Never, I never leave home*’ (ID 6, 3 months). Another participant stated that he is not able to practice activities outdoors nor explore the neighbourhood or the city due to severe physical problems: ‘*Yes (I mostly spend my time at home) because physically I’m not fine, now everything has changed*’ (ID 8, 4 months).

Some of the participants declared that they have nothing to do at home and that they struggle to fill time with activities: ‘*I have nothing to do […] You won’t believe me, I go to bed (after breakfast) and I stay there until 2.30 pm*’ (ID 6, 3 months). One participant discussed the fact that he spent a considerable amount of time online using social networking sites: ‘*I spend a lot of time on Facebook as I am an estranged […]*’ (ID 5, 5 months).

Moreover, one participant expressed a strong intolerance towards her accommodation as she was suffering because of the long distance between her house and the city centre and the neighbourhood where her previous accommodation was located. She reported: ‘*I would only like to go out from these walls, at least two or three times a week, that’s it […] I want to leave this cage, I don’t want to spend the entire day reading continuously, the flat is small […]*’ (ID 13, 5 months).

#### 3.2.3. Going Outside: Using Resources from the Community

Participants described frequenting places in the neighbourhood and the broader community to satisfy daily and tangible needs. They frequent local grocery stores, newsstands, cafés, tobacco shops, and parks to shop, have some meals, or have a walk. Two participants reported that they visit the library to find some interesting books or to browse newspapers and magazines: ‘*I usually go to the library […] I read magazines that are on the third floor. Otherwise, I go to the second floor where the library is located, and in fact, right now I do have a couple of books from there and I borrow books*’ (ID 14, 3 months); ‘*Then sometimes I go to the library in the city centre […] if there is something that interests me I go there*’ (ID 2, 7 months).

In addition to frequenting community services to pursue instrumental goals, some people say they enjoy going to certain places to socialise and meet friends and acquaintances, as two participants reported: ‘*In the morning I usually go to the hospital to have a coffee […] I always meet my friends there, where I go to have coffee*’ (ID 4, 6 months); ‘*I bring (my friend) to the shop close to the hospital owned by another friend who can make an exceptional sandwich. I made him try it once and now every time we meet, we go together there to eat it*’ (ID 2, 7 months). One person said she enjoys returning to her old neighbourhood to visit the places she used to visit and see the people she used to know: ‘*So I go and I greet those at the bar, the woman at the flower shop, the other one of the bar, the one at the newsstand. I go and greet everybody*’ (ID 13, 5 months).

Another participant reported that he loves going to the bar for relational purposes, as he enjoys talking to the owners: ‘*In the evening, I go to some bars in the city centre. I highly recommend one in particular in the centre […] It is nice because even if I could not find anybody, there would always be owners I can turn to. I have always loved bars where I can bond with the owners*’ (ID 5, 5 months). Overall, participants generally used community services to fulfil basic (food and purchase) and relational needs to meet friends or acquaintances.

#### 3.2.4. Staying Outside: Using Homeless Services

Some participants claimed that they still rely on homeless services such as day centres and services distributing food, even though they now own an apartment. One participant said he visits the soup kitchen during mealtimes, as he recently moved into a new apartment without a kitchen: ‘*At 11:30 am I go and eat at the soup kitchen*’ (ID 1, 1 month). He additionally said that he does not like going out and that he only does it for work and to go to the soup kitchen: ‘*I don’t like going anywhere, just at home […] I only go to the soup kitchen, then I go home*’ (ID 1, 1 month).

Another individual reported that he relies on the day centre for meals and typically collects food there to reheat for dinner at home: ‘*In the evening I stay home and sometimes it happens that at the day centre they give me food, for example, today they gave me this quiche so I can warm it up in the oven*’ (ID 4, 6 months). For some people, services remain essential to satisfy basic needs, such as food, due to inadequate housing conditions and the maintenance of previous daily routines.

## 4. Discussion and Implications

This research aimed to explore the daily lives of people with a history of homelessness living in public houses, focusing on their social relationships and the places where they frequent.

Participants reported having a daily routine, given that they met friends and family members and keep contact with others in the community. Many of them have maintained a connection with homeless services not only for material aid but mostly to meet other people and social service providers with whom they socialise. During the day, participants attended community places to meet basic needs (e.g., grocery shopping) but also to cultivate hobbies (e.g., going to the library) or see other people. These aspects can be traced back to the social dimension of community integration [23]; however, the most meaningful social relationships seem more anchored in homeless services than in the community.

Considering these results, some considerations are important. Firstly, people never talk about relationships within their homes. Participants recounted meeting friends or people around the city but did not talk about the home as a place of relationship, unlike findings in other studies with other populations [4]. The reason could be that people had recently purchased a new home and thus needed to feel the home as their place of identity through personalisation of surroundings, an essential component of the psychological home [5].

Moreover, these relationships seem to be superficial. People talk about meeting friends or people at bars or other places in the community but do not explicitly tell them about having emotional connections or doing something meaningful with them. They usually reported having phone calls, greetings, and walks with friends but did not mention meaningful relationships. As shown in Figure 1, half of the participants met acquaintances and friends, but only a few of them did anything with them.

Despite independent housing, participants continued to have social relationships with people from homeless services, especially with social service providers. From the participants’ words, it seems that they found a greater sense of safety, sociability, and warmth in homeless services than in their homes. Almost all participants (13 out of 14) asked for help from services when needed. These services are not only instrumental to meet basic needs (e.g., having food or looking for a job) but also to meet people they trust, to whom they feel connected.

Finally, as demonstrated in previous studies, some people prefer not to have relationships and prefer to stay home alone due to aspects of their character or past life events [19,20].

People create daily routines by going around the city, meeting people in bars, or providing homeless services, almost because otherwise, they would have nothing else to do. Participating in community or homeless service activities helps fulfil important relational goals to socialise and establish weekly routines.

As demonstrated in another study, people sometimes prefer to stay at home as the main place where they spend the day [28]. The reason is that although they have found a place of peace and security, they remain at risk of living in isolation because they have nothing else to do. The house is viewed in two ways. On the one hand, the house is a place of refreshment and relaxation and where one can enjoy the quiet after work. On the other hand, the home is experienced as a place to stay because they do not know what else to do or where to go. A home is seen here as the place where privacy and self-determination are restored, consistent with past research on the role of home on ontological security [2]. However, given that they cannot choose the location of the new home, some people do not feel at ease in their new surroundings, either because they are unfamiliar with the environment or because they lack meaningful activities to keep them occupied during the day.

For some people, the lack of a job or meaningful activities seems to hinder (re)integration into the broader community and the restoration of satisfactory daily routines. This underlines the necessary but insufficient role of a home for people with more complex needs, such as those who have experienced homelessness [29]. Moreover, this is in line with other studies that emphasised the importance of meaningful activities for people’s recovery and well-being [30,31,32] and for giving people a daily routine and a sense of usefulness in society [25]. A recent study showed how everyday activities contribute to the transition from homeless to permanent homes. Moreover, the study highlighted the need for more personalised support to help people sustain their accommodation, for example, adapting the physical environment to make people feel at home and having a daily occupational routine [33]. On the basis of these results, we suggest recommendations for future practice. For instance, social service providers should help them personalise their space as they enter their new homes. Given that a psychological home includes a sense of identity related to place, people must perceive the physical environment as an expression of their identity [4]. In addition to the home, homeless service providers should assist people in cultivating social relationships at home and familiarising themselves with their neighbourhood, for example, through neighbourhood walks [20]. Finally, investment should be made in programs, such as Housing First, in which housing is considered a basic right but also flexible and ongoing support to a person is provided, supporting people to make choices about their recovery and daily living activities [34]. Thus, rather than having people enter services with the risk of never escaping homelessness and the associated stigma, social service providers should assist people in entering their new homes.

## 5. Limitations

Despite the valuable insights provided by the results, this study has some limitations. Firstly, some sociodemographic variables were not considered when exploring potential differences in the experiences of the participants. For example, there may have been differences in daily life experiences based on their personal background in terms of history of homelessness, neighbourhood where one lives, or gender. Future research can further explore this topic by considering other contextual aspects, such as the housing situation and the neighbourhood.

Secondly, changes in people’s daily lives must be monitored because some processes take time. During the writing of this article, we conducted interviews six months apart to explore whether people had changed their daily lives in terms of relationships, places they frequent, and life plans.

Finally, some participants may have been more reluctant to share details of their personal experiences in front of social service providers, with the risk of a social desirability bias. To mitigate this, the interviewers emphasised the importance of protecting the participants’ privacy and the fact that it would not influence their housing stability in any way because they had a regular housing contract.

## 6. Conclusions

Overall, this study contributes to understanding the daily lives of people living at home after a long period of homelessness. The results highlighted the importance of helping people cultivate social relationships and daily routines in their homes to help them escape homelessness. More efforts are warranted to help people feel a sense of psychological home, especially for those who have experienced homelessness for a long time. Particularly, research and practice should explore the mechanisms that help people experience their home as a place of identity and meaningful relationships.

## Figures and Tables

**Figure 1 behavsci-12-00416-f001:**
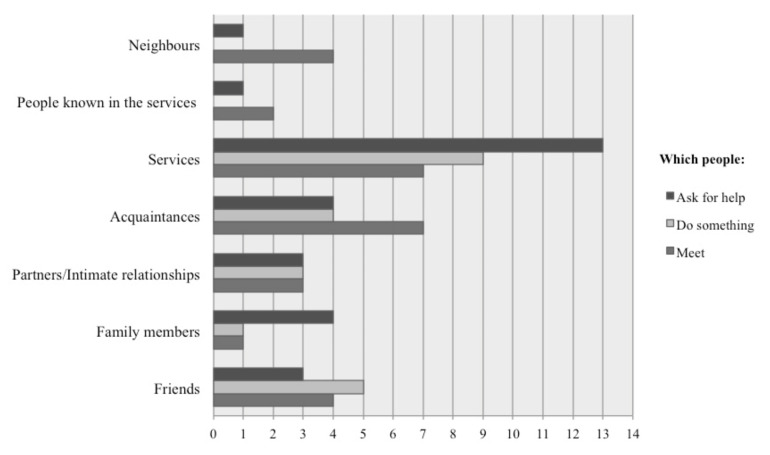
Frequency of responses to the question: ‘With whom do you meet/do something/get help?’.

**Figure 2 behavsci-12-00416-f002:**
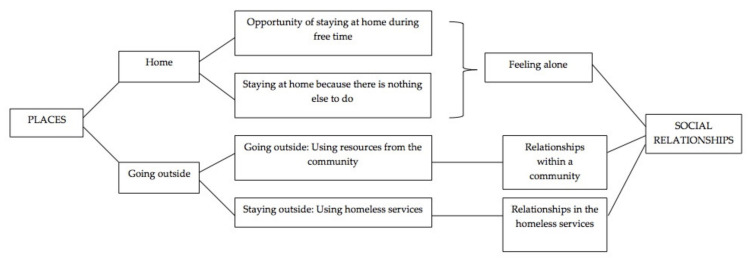
Generated themes concerning social relationships and places.

## Data Availability

The data that support the findings of this study are available from the corresponding author upon reasonable request.

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
