# Peer review of "In My New Home: The Daily Lives of People Living in Public Houses after a Long Period of Homelessness"

_behavsci, 2022, doi:10.3390/bs12110416_

Round 1

Reviewer 1 Report

This is an interesting study looking at the formerly homeless population in Padua, Italy and their experiences as they transition out of the homeless population.  I think this is an interesting topic, one that is demonstrates a gap in the academic literature.  My main issue with the paper is your English grammar, style, and prose.  There are significant editorial errors throughout the paper, making it difficult to read at times.  I found subject-verb disagreements, missing articles, unclear word choices and a few typographical errors.  I would strongly encourage you to have a native English speaker proofread your paper before you resubmit.  A few of the errors I wanted to point out because they significant impact the strength of your argument.  First, in line 10 of the abstract, you mention that "crucial role of the home for the person's well being is now well known."  I'm unsure if you wanted to say "not well known."  Second, in line 16-17 in the abstract, you state your findings "yield different and opposed themes."  You might want to expand on this idea slightly.  I did not understand what you mean here.  Third, in line 38, "low life expectancy" should probably be "a lower life expectancy."  In line 53, you mention "encouraging evidence."  I'm not sure "encouraging" is the correct word here.  Fifth, in line 111 you mention "semi-structured interviews."  Are these the "open ended questions" you mentioned earlier in the paragraph?  If so, I would encourage you to keep the terminology the same to help with readability.  If not, then you should discuss the open ended questions in this section.  There are several more examples of stylistic and editorial mistakes in the paper.  I would suggest you address these issues to make the paper more readable.  

Overall, I think you have a solid command of the academic literature on the topic.  I appreciate the organization of your literature review.  I think your interview questions logically follow your literature review.  You do an outstanding job discussing the results of your interviews and your conclusions logically follow your results and tie back to the literature review.  

Thank you for a very interesting article.  If you address the style, grammar and usage issues in the paper, you will have an important contribution to the academic literature.  Best of luck with your revisions.  

Author Response

Reviewers’ and Editor’s Comments and Author Replies

Dear Editor and Reviewers,

Thank you very much for the opportunity to revise and resubmit our manuscript entitled: “In my new home: the daily lives of people living in public houses after a long period of homelessness”. We would like to thank both the editor and the reviewers for their valuable input, which we believe helped us enhance the quality of the manuscript.

We have addressed all comments and suggestion made by the reviewers and adjusted the manuscript accordingly. An English expert proofread the manuscript. If necessary we have a language editing certificate.

Below we present the changes we made in detail, following the same order the reviewers adopted when highlighting the issues. Below you will find our responses to the reviewers, namely comments and specific changes we made to the paper. Please find attached the revised version of the manuscript (with edits marked) together with our “Response to Reviewers’ comments”. We hope that the manuscript is now acceptable for publication, we are available for any other changes.

Academic Editor’s comments

The study is "small", but the theme is very significant and the participants difficult to reach. Although generalizations are not possible I find that the work provides insights for researchers, but also for social workers, with important practical implications. The method is well described, I would suggest the introduction of figures/ tables to facilitate the reading of the results, but I refer to the peer-reviewers any requests in this regard.

Authors’ reply.

Thank you so much, we appreciate that you acknowledged our efforts.

Because we followed Braun & Clarke's (2006) directions for our qualitative research, we did not consider sample size as a limitation because the intent of the research was not to generalize the results but to explore in depth the daily lives of a group of people in a specific context. However, we are available to supplement the discussion of limitations.

Overall, an English expert proofread the manuscript. If necessary we have a language editing certificate. We have added Figure 2, which summarizes the themes identified. We changed one word in the title.

Reviewer 1

Comments to the Authors:

Comment 1. This is an interesting study looking at the formerly homeless population in Padua, Italy and their experiences as they transition out of the homeless population. I think this is an interesting topic, one that is demonstrates a gap in the academic literature. My main issue with the paper is your English grammar, style, and prose. There are significant editorial errors throughout the paper, making it difficult to read at times. I found subject-verb disagreements, missing articles, unclear word choices and a few typographical errors. I would strongly encourage you to have a native English speaker proofread your paper before you resubmit. A few of the errors I wanted to point out because they significant impact the strength of your argument.

Authors’ reply #1.

Thank you for appreciating our study. We are sorry that there were so many errors, now an English expert has proofread the paper.

Comment 2. First, in line 10 of the abstract, you mention that "crucial role of the home for the person's well being is now well known."  I'm unsure if you wanted to say "not well known."

Authors’ reply #2.

We rephrased the sentence: “Although the literature has revealed the crucial role of a home in a person’s well-being, few studies have explored the daily lives of people who transition from homeless services to an independent home.”

Comment 3. Second, in line 16-17 in the abstract, you state your findings "yield different and opposed themes." You might want to expand on this idea slightly. I did not understand what you mean here.

Authors’ reply #3.

We rephrased the sentence: “Several themes concerning social relationships and places were identified.”

Comment 4. Third, in line 38, "low life expectancy" should probably be "a lower life expectancy." 

Authors’ reply #4.

Thanks, we have corrected the error.

 Comment 5. In line 53, you mention "encouraging evidence."  I'm not sure "encouraging" is the correct word here. 

Authors’ reply #5.

We rephrased the sentence: “Despite the evidence supporting the importance of housing, housing alone is not always sufficient to address the issues associated with homelessness, such as social exclusion.”

Comment 6. Fifth, in line 111 you mention "semi-structured interviews."  Are these the "open ended questions" you mentioned earlier in the paragraph?  If so, I would encourage you to keep the terminology the same to help with readability. If not, then you should discuss the open ended questions in this section. There are several more examples of stylistic and editorial mistakes in the paper. I would suggest you address these issues to make the paper more readable. 

Authors’ reply #6.

We restructured the "procedure" section by explaining that the interviews included closed questions (self-report questionnaires) and open-ended questions. We removed the reference to semi-structured interviews.  

Comment 7. Overall, I think you have a solid command of the academic literature on the topic. I appreciate the organization of your literature review. I think your interview questions logically follow your literature review. You do an outstanding job discussing the results of your interviews and your conclusions logically follow your results and tie back to the literature review. 

Thank you for a very interesting article. If you address the style, grammar and usage issues in the paper, you will have an important contribution to the academic literature.  Best of luck with your revisions.

Authors’ reply #7.

Thank you so much, we appreciate that you acknowledged our efforts. We hope that the manuscript is now acceptable for publication.

Reviewer 2 Report

Correctly written paper concerning a current and interesting topic by an experienced team of authors.

Major remarks.

As a reviewer and author of other publications, I am used to papers slightly more based on statistical analyzes. Perhaps that is why I notice a certain methodological weakness in the proposed article. I miss here (strongly) something like "study population subsection"

Description – line 88 – “…people who had experience homelessness for at least 2 years…“ it is too laconic and insufficient. Who were they? I am in adherent of using the ETHOS by FEANTSA classification (https://www.feantsa.org/en/toolkit/2005/04/01/ethos-typology-on-homelessness-and-housing-exclusion), it makes future comparisons much easier. I understand that it may not be possible to implement it into your study at this time (after the end of the study), but you should at least categorize and describe them (the homeless do not constitute some separate race but derive from society at the margin of which they live).

I miss the Limitations subsection as well. All the paper is based on a relatively small database of people who were somehow pre-selected by social services, flats were not distributed to the homeless at random - where they?

And two minor remarks regarding the selection of references.

Position 6 is weak, it may remain, but should be backed up by something more recognizable.

Position 8 – In my opinion below citation fits better.

Romaszko, J., Cymes, I., Dragańska, E., Kuchta, R. and Glińska-Lewczuk, K., 2017. Mortality among the homeless: Causes and meteorological relationships. PloS one, 12(12), p.e0189938.

Author Response

Reviewers’ and Editor’s Comments and Author Replies

Dear Editor and Reviewers,

Thank you very much for the opportunity to revise and resubmit our manuscript entitled: “In my new home: the daily lives of people living in public houses after a long period of homelessness”. We would like to thank both the editor and the reviewers for their valuable input, which we believe helped us enhance the quality of the manuscript.

We have addressed all comments and suggestion made by the reviewers and adjusted the manuscript accordingly. An English expert proofread the manuscript. If necessary we have a language editing certificate.

Below we present the changes we made in detail, following the same order the reviewers adopted when highlighting the issues. Below you will find our responses to the reviewers, namely comments and specific changes we made to the paper. Please find attached the revised version of the manuscript (with edits marked) together with our “Response to Reviewers’ comments”. We hope that the manuscript is now acceptable for publication, we are available for any other changes.

Academic Editor’s comments

The study is "small", but the theme is very significant and the participants difficult to reach. Although generalizations are not possible I find that the work provides insights for researchers, but also for social workers, with important practical implications. The method is well described, I would suggest the introduction of figures/ tables to facilitate the reading of the results, but I refer to the peer-reviewers any requests in this regard.

Authors’ reply.

Thank you so much, we appreciate that you acknowledged our efforts.

Because we followed Braun & Clarke's (2006) directions for our qualitative research, we did not consider sample size as a limitation because the intent of the research was not to generalize the results but to explore in depth the daily lives of a group of people in a specific context. However, we are available to supplement the discussion of limitations.

Overall, an English expert proofread the manuscript. If necessary we have a language editing certificate. We have added Figure 2, which summarizes the themes identified. We changed one word in the title.

Reviewer 2

Comments to the Authors:

Comment 1. Correctly written paper concerning a current and interesting topic by an experienced team of authors.

Major remarks. As a reviewer and author of other publications, I am used to papers slightly more based on statistical analyzes. Perhaps that is why I notice a certain methodological weakness in the proposed article. I miss here (strongly) something like "study population subsection"

Authors’ reply #1.

Thank you so much. We added a "Participants" section specifying some socio-demographic characteristics of the participants.

Comment 2. Description – line 88 – “…people who had experience homelessness for at least 2 years…“ it is too laconic and insufficient. Who were they? I am in adherent of using the ETHOS by FEANTSA classification (https://www.feantsa.org/en/toolkit/2005/04/01/ethos-typology-on-homelessness-and-housing-exclusion), it makes future comparisons much easier. I understand that it may not be possible to implement it into your study at this time (after the end of the study), but you should at least categorize and describe them (the homeless do not constitute some separate race but derive from society at the margin of which they live).

Authors’ reply #2.

We specified this according to the ETHOS classification by explaining where people lived before entering the home: “Participants had experienced homelessness for 10 years (range: 4–27, SD= 6.5). Before housing, seven lived in a public shelter, five in a group home, two in a street, or an abandoned house. According to the European Typology of Homelessness and Housing Exclusion (ETHOS) [26], participants were roofless (without a shelter of any kind, sleeping rough) or houseless (with a place to sleep, but temporary in institutions or shelters). At the time of the study, people lived in their house for one to a maximum of 11 months.”

Comment 3. I miss the Limitations subsection as well. All the paper is based on a relatively small database of people who were somehow pre-selected by social services, flats were not distributed to the homeless at random - where they?

Authors’ reply #3.

We have added a "Limitations" section. Moreover, we specified in the Procedure section that: “The municipality of Padua published its annual call for public housing allocation. Housing is allocated on the basis of the score obtained in the ranking list and compliance with specific requirements (e.g. low income). In 2021, 27 people experiencing homelessness won the ranking list and became homeowners.” 

Comment 4. And two minor remarks regarding the selection of references.

Position 6 is weak, it may remain, but should be backed up by something more recognizable.

Position 8 – In my opinion below citation fits better.

Romaszko, J., Cymes, I., Dragańska, E., Kuchta, R. and Glińska-Lewczuk, K., 2017. Mortality among the homeless: Causes and meteorological relationships. PloS one, 12(12), p.e0189938

Authors’ reply #4.

Thank you for the suggestions. We have replaced the references in position 6 and 8. We have also included a recent study in the discussion: “A recent study showed how everyday activities contribute to the transition from homeless to permanent homes. Moreover, the study highlighted the need for more personalised support to help people sustain their accommodation, for example, adapting the physical environment to make people feel at home and having a daily occupational routine [33].”

Round 2

Reviewer 1 Report

Thank you for addressing my comments from the first submission.  This version of the manuscript is vastly improved from the first version.  I think this is an interesting article with the potential to fill a void in the academic literature.  My only minor comment deals with your description of "services" (starting in line 126).  I think I understand what "services" means, but maybe include some examples of specific services to make your discussion clearer.  For example, you mention that people "hang out at homeless services " (Line 216).  Which homeless services do they hang out at?  This is not a major problem, just something that would help clarify and allow readers to understand how the network works in Padua.  Overall, I really enjoyed your article and I learned a lot from reading it.  Thank for you the opportunity to learn and best of luck in the future.  

Author Response

Thank you so much, we appreciate that you acknowledged our efforts. Thanks for the suggestion, we specified along the text the services we refer to. Specifically the services that people continue to attend are day centres and services that distribute food. We have highlighted the changes in yellow.

Reviewer 2 Report

I have no further remarks.

Author Response

Thank you so much!